# Kaempferol Alleviates Hepatic Injury in Nonalcoholic Steatohepatitis (NASH) by Suppressing Neutrophil-Mediated NLRP3-ASC/TMS1-Caspase 3 Signaling

**DOI:** 10.3390/molecules29112630

**Published:** 2024-06-03

**Authors:** He Yang, Dandan Li, Guolan Gao

**Affiliations:** Medical School, University of Chinese Academy of Sciences, Beijing 101408, China

**Keywords:** NASH, NLRP3, ASC/TMS1, Caspase 3, apoptosis

## Abstract

Background: Non-alcoholic fatty liver disease (NAFLD) is a significant hepatic condition that has gained worldwide attention. Kaempferol (Kae), renowned for its diverse biological activities, including anti-inflammatory, antioxidant, anti-aging, and cardio-protective properties, has emerged as a potential therapeutic candidate for non-alcoholic steatohepatitis (NASH). Despite its promising therapeutic potential, the precise underlying mechanism of Kae’s beneficial effects in NASH remains unclear. Therefore, this study aims to clarify the mechanism by conducting comprehensive in vivo and in vitro experiments. Results: In this study, a murine model of non-alcoholic steatohepatitis (NASH) was established by feeding C57BL/6 female mice a high-fat diet for 12 weeks. Kaempferol (Kae) was investigated for its ability to modulate systemic inflammatory responses and lipid metabolism in this model (20 mg/kg per day). Notably, Kae significantly reduced the expression of NLRP3-ASC/TMS1-Caspase 3, a crucial mediator of liver tissue inflammation. Additionally, in a HepG2 cell model induced with palmitic acid/oleic acid (PA/OA) to mimic NASH conditions, Kae demonstrated the capacity to decrease lipid droplet accumulation and downregulate the expression of NLRP3-ASC/TMS1-Caspase 3 (20 µM and the final concentration to 20 nM). These findings suggest that Kae may hold therapeutic potential in the treatment of NASH by targeting inflammatory and metabolic pathways. Conclusions: These findings suggest that kaempferol holds potential as a promising therapeutic intervention for ameliorating non-alcoholic fatty liver disease (NAFLD).

## 1. Introduction

Non-alcoholic steatohepatitis (NASH) is a widespread liver disorder typified by the infiltration of inflammatory cells and excessive accumulation of lipid droplets within hepatocytes. This condition poses a substantial health concern, warranting the diligent investigation of innovative therapeutic approaches [1]. The spectrum of non-alcoholic fatty liver disease (NAFLD) encompasses two distinctive presentations. The initial stage, known as steatosis, involves the benign accumulation of lipids within the hepatic cells. However, the advanced form, referred to as NASH, is characterized by the presence of inflammation and hepatocellular injury. This differentiation is pivotal for comprehension of the disease’s progression and the formulation of targeted treatment approaches [2]. The etiology of NASH remains enigmatic, with its development attributed to a multifactorial interplay, including genetic predispositions related to lipid metabolism, oxidative stress, and inflammatory processes [3]. Approximately one-quarter of the global population is affected by NAFLD, with NASH being a significant subset [4]. This condition is a foremost contributor to cirrhosis and hepatocellular carcinoma [5]. Regional prevalence rates of NAFLD exhibit variability, with higher incidences reported in Europe and the United States, where it exceeds 20%, while in the Asia–Pacific region, the rates span from 5% to 32% [6]. Estimates project a staggering 63% rise in the number of NASH cases by 2030. To combat the root causes and progression of this disease, therapeutic interventions under consideration for NAFLD focus on compounds with anti-inflammatory and antioxidant capabilities [7]. Additionally, ongoing research is exploring the potential of agents that can mitigate lipid accumulation as a strategy for managing NAFLD [8].

The NOD-like receptor protein 3 (NLRP3) inflammasome is a critical signaling complex activated in response to liver injury and fibrosis progression, making it a compelling therapeutic target [9]. This multi-component protein assembly is composed of three main elements: (1) NLRP3, functioning as a central interaction platform, (2) ASC (apoptosis-associated speck-like protein containing a CARD), serving as an adapter protein, and (3) pro-caspase-1, a protease enzyme involved in inflammatory processes [10]. Activation of nuclear factor-κB (NF-κB) upon injury signals triggers the production of NLRP3 and pro-interleukins, such as pro-IL-1β, leading to inflammasome activation and IL-1β release, exacerbating inflammation and cell death. ASC/TMS1 plays a central role in both apoptosis and inflammation, coordinating caspase-mediated apoptosis through interactions with caspase-8 and -9, depending on the cellular context [11]. It promotes caspase-8-driven BID proteolysis, activating the mitochondrial apoptotic pathway independently of FADD in certain cell types. ASC/TMS1 also facilitates BAX migration to mitochondria, triggering BAX-dependent apoptosis along with caspase-9, -2, and -3 activation [12,13,14]. As a mediator, ASC/TMS1 contributes to inflammasome activation by recruiting caspase-1 to sites where pattern recognition receptors, like NLRP2 and NLRP3, are localized [15]. Bcl-2, known for its anti-apoptotic properties, regulates cell death in various cell systems, including lymphohematopoietic and neural cells, by modulating mitochondrial membrane permeability [16,17]. PCNA, a vital protein in DNA replication, repair, and cell cycle control, interacts with similar molecules to support efficient DNA processing, serving as a marker of cellular proliferation [18,19,20,21]. These interconnected mechanisms highlight the complexity of NASH pathogenesis and the potential targets for therapeutic intervention.

Throughout history, humans have utilized the therapeutic benefits of natural products. These compounds have the potential to alleviate a range of ailments. It is important to use a clear and objective perspective when discussing their properties [22]. In the realm of medical science, we found the kaempferol (Kae), a flavonoid of significant biological value found abundantly in numerous botanical sources [23]. Extensive research has demonstrated Kae’s diverse therapeutic properties, encompassing anti-inflammatory and antioxidant capabilities. Furthermore, it has shown promising potential in combating the aging process and providing protection against cardiovascular diseases [24,25,26]. Kae also has prowess in mitigating the risk of liver failure induced by the harmful lipopolysaccharide–galactosamine, achieved through the moderation of the deleterious endoplasmic reticulum stress (ERS) [27]. Furthermore, the influence of Kae on the regulation of ERS reveals a complex ambivalence—a delicate balance between promoting apoptosis in malignant cells while also seeking to prevent the same fate for non-tumor cells [28]. However, despite these insights, the mechanisms by which Kae provides relief for NASH remain unclear and await further revelation.

Our study revealed that mice consuming a HFD exhibited liver problems linked to the NLRP3-ASC/TMS1-Caspase 3 pathway. Cells treated with PA/OA showed fat accumulation. Markers of liver inflammation, such as neutrophil enzymes MPO and NE, were also observed. By introducing Kae, we observed a decline in the expression of the NLRP3-ASC/TMS1-Caspase 3 pathway and a decrease in inflammatory substances. Consequently, Kae demonstrated potential in mitigating liver damage associated with HFD. In essence, our findings position Kae as a promising therapeutic agent for addressing NASH, offering a ray of hope for improved treatment outcomes.

## 2. Results

### 2.1. The Impact of Kaempferol on HepG2 Cells

In line with prior research, we established a concentration gradient of Kae (Figure 1A) and discovered that HepG2 cells exhibited the maximum enhancement in functionality with the addition of 20 µM Kae. To mimic the high-fat conditions seen in HFD in the cellular context, we treated HepG2 cells with PA/OA. Our investigation focused on the impact of Kae on inflammation-related gene expression in these cells. By supplementing the HFD-like conditions with Kae, we observed a decrease in the expression of pro-inflammatory genes, such as IL-1β, IL-6, and TNF-α (Figure 1B). Conversely, the expression of the anti-inflammatory gene IL-10 was increased in these cells. These findings imply that Kae might have a protective role against NASH at the cellular level, as it helps to modulate the inflammatory response.

### 2.2. Immunofluorescence Assay for Assessing Proliferative Function in HepG2 Cell Line following Kaempferol Treatment

In addition to assessing the inflammatory gene expression, we also examined the expression of proteins related to anti-apoptosis, proliferation, and anti-inflammation using immunofluorescence. In cells subjected to a HFD, the expression of Bcl-2 (Figure 2A,D), an anti-apoptotic protein, and PCNA (Figure 2B,E), a marker of cell proliferation, was increased and comparable to that of the NC group. This suggests that Kae supplementation promotes cell survival and growth. On the other hand, the expression of p65 (Figure 2C,F), a protein involved in inflammation signaling, was reduced in the Kae-treated group compared to the HFD group.

### 2.3. The Effect of Kaempferol on Lipid Metabolism in NASH Model Mice

This study delves into the direct impact of Kae on NASH in mice models. Initially, we monitored the body weight of the animals, and it was observed that Kae-treated mice had a reduced body weight compared to the HFD group, resembling the normal control (NC) group (Figure 3A). Furthermore, the liver weight of the Kae group was closer to the NC group, indicating a healthier liver state (Figure 3B). Considering the liver’s role in metabolism, we also assessed the weight of perirenal and mesenteric fat, and the Kae group displayed improved outcomes (Figure 3C). In terms of biochemical markers, Kae treatment led to lower levels of total cholesterol (TC), triglycerides (TG), and low-density lipoprotein (LDL), which are indicative of fat accumulation and lipid metabolism disturbances, though still slightly higher than the NC group. The high-density lipoprotein (HDL) levels, which are protective, were increased, suggesting Kae’s potential in addressing metabolic disorders (Figure 3D). Regarding liver injury, the levels of aspartate aminotransferase (AST) and alanine aminotransferase (ALT) were marginally elevated in the Kae group, significantly less than the HFD group, implying minimal liver tissue damage (Figure 3D). Overall, these findings support the notion that Kae treatment can positively influence NASH symptoms by improving metabolic parameters and reducing liver injury.

### 2.4. The NLRP3-ASC/TMS1/Caspase-3 Pathway Initiated Liver Injury in NASH Mice

To assess liver injury, we examined histological alterations using hematoxylin and eosin (HE) staining. In the mice fed a HFD, liver cells displayed substantial lipid accumulation and signs of cellular damage. However, upon Kae treatment, these morphological changes were notably reduced (Figure 4A), indicating an improvement in liver health. To understand the underlying cause of liver injury, we focused on the interplay between lipid metabolism, mitochondria, and the chronic inflammation characteristic of NASH. We analyzed the expression of NLRP3-ASC/TMS1-Caspase 3, a pathway linked to inflammation and mitochondrial apoptosis. The HFD group exhibited increased expression of these proteins, while Kae treatment led to a decrease in their expression, reverting the trend (Figure 4B–E). 

### 2.5. The Function of Kaempferol Supplements on Inflammation in a NASH Mouse Model

As previously discussed, we extended our investigation to assess pro-inflammatory factors in liver tissue and serum, particularly focusing on HE stains and the NLRP3 pathway protein expression. Our results reveal the presence of inflammation in the high-fat diet (HFD) group, as demonstrated by the elevated levels of pro-inflammatory cytokines, including IL-1β, IL-6, and TNF-α, both in liver tissue and serum samples when compared to the normal control (NC) and Kae-treated groups (Figure 5A–C,E,F). This overexpression indicates an inflammatory response in the HFD group. Conversely, the anti-inflammatory cytokine IL-10 was found to be downregulated in the HFD group (Figure 5D,H), suggesting a reduced capacity to counteract inflammation. This decrease in IL-10 levels further supports the inflammatory state in the HFD group and highlights the importance of restoring the balance between pro-inflammatory and anti-inflammatory factors.

### 2.6. The Activation of MPO and NE Overexpression Caused Injury in NASH Mice

To gain deeper insights into the cause of liver injury, we performed immunohistochemistry (IHC) analyses on liver tissue samples, focusing on inflammation-related markers such as myeloperoxidase (MPO) (Figure 6A), neutrophil elastase (NE) (Figure 6B), IL-1β (Figure 6C), and TNF-α (Figure 6D). The presence of enlarged lipid droplets in the liver tissue made it challenging to measure the average grayscale values accurately. However, it was evident that MPO, NE, IL-1β, and TNF-α were overexpressed in the HFD group compared to the NC group, indicating chronic inflammation. The Kae group displayed a marked improvement over the HFD group, with a significantly reduced active area of these inflammatory proteins. To further understand the interconnectedness of these processes, we also analyzed these protein expressions in perivisceral adipose tissue, given its close relationship with the liver. The HFD group showed the largest lipid droplets and the highest expression of inflammatory proteins in the liver tissue (Figure 6E–G).

### 2.7. Effect of Kaempferol on Proliferation in Liver Tissue of NASH Model Mice

The study aimed to investigate how kaempferol can protect the liver from chronic inflammation. Immunohistochemistry was used to assess the expression of Bcl-2 (Figure 7A), PCNA (Figure 7B), and P65 (NF-κB) (Figure 7C). The results show that BCL-2 is overexpressed in the Kae group compared to the NC and HFD groups, indicating that Kae treatment could upregulate anti-apoptosis function. PCNA, which represents the function of proliferation, is expressed at higher levels in both the NC and Kae groups but is exhausted in the HFD group. The proinflammatory marker P65 (NF-κB) is significantly expressed, albeit slightly, in Kae compared to the NC group.

## 3. Discussion

NAFLD has become a major global health concern, with its incidence rapidly rising across the world [29,30]. The situation has worsened, as the number of NAFLD cases has surged by an alarming 29.1% since 2016, reaching a staggering 246.33 million individuals [31]. Among its subtypes, NASH, characterized by inflammation, is a critical stage that can progress to cirrhosis and liver failure, often necessitating liver transplantation. Unfortunately, NASH is often underdiagnosed and underestimated in clinical settings, despite its profound implications for patient health [32]. As the awareness of NAFLD’s extensive impact on well-being grows, the medical community must intensify its efforts in researching, diagnosing, and managing this condition [6,33]. The increasing interest in NAFLD research is justified, considering that NASH is now the primary reason for liver transplantation in the United States, emphasizing the exigency for innovative and effective treatment options [32]. To combat this escalating health crisis, it is crucial for researchers and healthcare professionals to collaborate on developing novel diagnostic tools and therapeutic interventions. This includes exploring natural compounds, such as kaempferol, which have shown promise in mitigating inflammation and liver injury associated with NASH. By focusing on these strategies, the medical community can work towards reducing the burden of NAFLD and improving the lives of millions affected by this disease.

Kaempferol (Kae), a naturally occurring flavonoid found in various fruits and vegetables, including *Kaempferol galanga* L., possesses a remarkable range of health-promoting properties. It has been shown to have antibacterial, anti-inflammatory, antioxidant, and even antitumor effects, making it a versatile compound with potential applications in multiple health conditions [34,35,36]. Research has consistently demonstrated kaempferol’s ability to effectively lower serum levels of ALT, LDL, and TC, as well as TG, lipid droplets, and the infiltration of inflammatory cells in the liver, thereby contributing to improved liver function [37]. This compound’s capacity to alleviate liver damage caused by harmful fatty acids like palmitic acid (PA) and oleic acid (OA) has been well-documented over time. These findings have been reinforced in in vitro studies using NASH cell models, particularly the HepG2 cell line, which underscores its potential as a therapeutic agent in the treatment of non-alcoholic steatohepatitis (NASH) [38]. Given its multifaceted therapeutic potential and proven efficacy in mitigating liver injury, kaempferol (Kae) presents an exciting opportunity for the development of novel treatments for NASH and other liver-related disorders. As the global health burden of NAFLD continues to rise, the exploration and utilization of natural compounds like kaempferol hold great promise for improving patient outcomes and addressing this pressing health issue.

In our previous research, we studied the protective effects of kaempferol (Kae) against non-alcoholic steatohepatitis (NASH) in a high-fat diet (HFD)-induced mouse model. In our latest study, we focused on the liver’s NLRP3-ASC/TMS1-Caspase 3 pathway, a critical component of inflammation, and its modulation by Kae. Our results show a marked decrease in the expression of these proteins in Kae-treated mice, contrasting with the upregulated expression observed in HFD-induced NASH mice. This suggests that Kae may effectively suppress inflammation by targeting this pathway. We also examined the role of neutrophil-induced inflammation, as indicated by MPO and NE expression. Our analysis reveals a substantial increase in MPO and NE, particularly MPO, in the liver and perivisceral adipose tissue of NASH mice, emphasizing their involvement in the inflammatory process. To strengthen our in vivo findings, we conducted in vitro experiments using HepG2 cells and induced steatosis with palmitic acid (PA) and oleic acid (OA). Kae treatment in these steatotic cells led to a suppression of NLRP3-ASC/TMS1-Caspase 3 expression, aligning with our observations in living organisms. In summary, our comprehensive study underscores kaempferol’s potential as a therapeutic agent in NASH by dampening liver injury and inflammation. These findings contribute to the understanding of NASH pathogenesis and offer promising avenues for the development of innovative treatment approaches. Further research is warranted to fully exploit the therapeutic potential of Kae and to translate these findings into clinical practice.

Inflammasomes, the vigilant guardians of the body, spring into action upon sensing any signs of danger, both from external pathogens and internal cellular stress. The NLRP3 inflammasome, a masterful responder to a myriad of stimuli, is a critical component of the innate immune system, defending the body against a vast array of threats [39,40]. NLRP3 acts as a watchful sentry, detecting both foreign invaders and signs of cellular distress, triggering the assembly of the NLRP3 inflammasome, a sophisticated molecular machine that includes NLRP3, caspase-1, and ASC [41]. NLRP3, the sensor protein, perceives danger signals, while caspase-1 executes inflammatory responses. ASC, the indispensable connector, binds these components together, forming a functional unit that initiates a coordinated immune response. Caspase-1 activation leads to the release of pro-inflammatory cytokines IL-1β and IL-18, triggering inflammation and pyroptosis, a programmed cell death essential for ridding the body of infected or damaged cells [42,43,44,45,46,47,48]. Caspase-3, a cysteine aspartate protease, plays a central role in apoptosis, ensuring a controlled and orderly cell death process that maintains cellular balance and eliminates unhealthy cells [49]. ASC, in its multifaceted role, not only activates the inflammasome but also participates in mitochondrial apoptotic pathways. It facilitates the activation of caspase-8, which in certain cell types, independently of FADD, matures BID. ASC also directs the translocation of BAX, orchestrating BAX-dependent apoptosis, and activates caspase-9, -2, and -3, ensuring a precise execution of the apoptotic process [50,51]. In this complex dance of life and death, ASC and caspase proteins work in harmony to maintain cellular homeostasis, eliminating threats and preserving the integrity of the organism. Understanding the intricate mechanisms of the NLRP3 inflammasome and its associated components is crucial for developing targeted therapies against inflammatory and autoimmune diseases.

In the conducted research, the administration of kaempferol (Kae) led to a remarkable restoration of equilibrium within the liver tissues of mice with non-alcoholic steatohepatitis (NASH). The observed decrease in the expression of NLRP3-ASC/TMS1-Caspase 3 proteins suggests that Kae effectively modulates the inflammatory response, contributing to the alleviation of liver injury. This outcome aligns with the hypothesis that the NLRP3-ASC/TMS1-Caspase 3 pathway, a critical component of the NLRP3 inflammasome, is downregulated in the NASH condition. These findings not only validate the role of the NLRP3 inflammasome in NASH pathogenesis but also highlight the potential of Kae as a therapeutic intervention. By targeting the NLRP3-ASC/TMS1-Caspase 3 pathway, Kae appears to mitigate liver damage and inflammation, offering a promising avenue for the treatment of NASH. This study underscores the importance of further research to explore the molecular mechanisms behind Kae’s action and to develop targeted therapies that could effectively combat NASH and its associated complications. In conclusion, the study’s results provide compelling evidence for the therapeutic potential of Kae in NASH treatment, as it appears to restore balance within the liver by modulating the NLRP3 inflammasome pathway. This discovery could pave the way for innovative approaches to managing this increasingly prevalent liver disease.

The intricate dance of immunity and inflammation relies heavily on cytokines, including interleukins, tumor necrosis factor (TNFα), and chemokines, which coordinate a comprehensive response to various stimuli [52]. TNFα and IL-6, in particular, act as key regulators, maintaining immune balance and guiding appropriate immune reactions. They play a pivotal role in numerous inflammatory conditions, demonstrating their importance in the body’s defense mechanisms [53,54]. Cellular models have shed light on Kae’s regulatory role in apoptosis, contributing to liver homeostasis restoration. This finding holds promise for developing innovative treatments for NASH and other inflammatory liver diseases. In our study, Kae was administered in both the NASH mouse model and a steatosis cell model induced by palmitic acid (PA) and oleic acid (OA). Adopting a formal academic tone with precise language, we observed a decrease in inflammatory factors and a transformation of the NLRP3-ASC/TMS1 pathway. Kae inhibited the caspase pathway, and the pro-inflammatory NF-κB, along with the proliferation marker PCNA, were suppressed.

In this research, Kae emerges as a harmonious regulator, delicately balancing inflammation and apoptosis by modulating NLRP3-ASC/TMS1-Caspase 3 expression. Our intervention offers a beacon of hope for liver injury resolution. However, we recognize the limitations of our study, having used C57BL/6 mice as models, and one part of our study was a retrospective analysis of clinical patients, and the other part was an animal study. According to our clinical data, menopausal women are suffering from metabolic disorder syndrome-related function, so we choose older females as our model animals. While these models provide valuable insights, caution is warranted when translating our findings to the human context, as previous research has emphasized the differences in liver cell proliferation between mice and humans [55,56]. This underscores the need for further studies to validate our results in human models and to fully understand the implications of Kae’s therapeutic potential in the treatment of liver diseases.

In the field of bioinformatics, the function of NAFLD demonstrates a more complex role. The insulin, peroxisome proliferator-activated receptor (PPAR), p53, and mitogen-activated protein kinase (MAPK) signaling pathways form a complex web that plays a pivotal role in the development of NAFLD and its progression to NASH [57]. These pathways, with their intricate interplay, hold the key to understanding the disease’s pathogenesis. In our future endeavors, we aim to delve into the depths of these pathways, examining their roles in the progression of NAFLD and NASH. Our research will focus on deciphering the alignment between kaempferol (Kae) interventions and these critical pathways. By exploring the communication between Kae and these signaling cascades, we hope to uncover new insights into the molecular mechanisms underlying Kae’s therapeutic potential. The insulin pathway, central to glucose metabolism, is often disrupted in NAFLD, leading to insulin resistance. PPARs, on the other hand, regulate lipid metabolism and inflammation, while the p53 pathway is involved in cellular stress response and apoptosis. The MAPK pathway, a master regulator of cellular processes, plays a role in inflammation, cell growth, and differentiation. Understanding how Kae interacts with these pathways could provide crucial information for developing targeted therapies for NAFLD and NASH. Our journey will be guided by the pursuit of knowledge, as we seek to unravel the mysteries that lie within these pathways and their relationship with Kae. By shedding light on these intricate connections, we aim to contribute to the development of more effective and personalized treatments for patients suffering from NAFLD and NASH.

## 4. Materials and Methods

### 4.1. Animals and Experimental Design

Female C57BL/6 mice aged 12 months and weighing 20–25 g were procured from Vital River Laboratory Animal Technology in Beijing, China. The mice were housed in a specific pathogen-free (SPF) environment with a temperature of 22 ± 2 °C and a 12-h light–dark cycle. They were provided with ad libitum access to food and water. After a week of acclimatization, 27 mice were randomly assigned to three groups, each consisting of 9 mice: (1) the nature control (NC) group, (2) the higher fatty acid-fed group (HFD) (details are shown in Table 1), (3) the higher fatty acid with kaempferol-fed group. Kaempferol, obtained from Sigma-Aldrich Co. in St. Louis, MO, USA, was dissolved in corn oil (2 mg Kae in 1 mL corn oil) at a dosage of 20 mg/kg per day for each mouse [58]. The NC group was fed a nature control diet, while the HFD and Kae groups were fed a high-fat diet with 60% fat energy. Additionally, the NC and HFD groups received equivalent quantities of corn oil. Kaempferol was administered continuously for 12 weeks. After a 12-h fasting period, the mice were anesthetized with sodium pentobarbital (150 mg per kg of body weight) and then euthanized by cervical dislocation [59]. After blood sampling, serum was collected from the retro-orbital plexus and stored at −80 °C. The liver was promptly excised and weighed for subsequent examination. The tissues were then stored at −80 °C for further analysis. This experiment was approved by the Institutional Animal Care and Use Committee (IACUC) of the Institute of Medical Laboratory Animals and Medical School, University of Chinese Academy of Sciences.

### 4.2. Serum Inflammation Factor and Metabolism Assay

Enzyme-linked immunosorbent assay (ELISA) was used to measure serum inflammation factor and metabolism markers, including total cholesterol (TC), triglycerides (TG), high-density lipoprotein (HDL), low-density lipoprotein (LDL), as well as aspartate aminotransferase (AST), alanine aminotransferase (ALT), interleukin-1β (IL-1β), interleukin-6 (IL-6), interleukin-10 (IL-10), and tumor necrosis factor-α (TNF-α). The ELISA kits were purchased from Elabscience (Wuhan, China) and used according to the manufacturer’s instructions.

### 4.3. Histological Analysis 

The liver tissues were dissected and washed with saline solution to remove any blood. They were then immersed in a 10% formalin buffer for 48 h to facilitate histological examination. After this period, the tissue sections were stained using the hematoxylin and eosin (H&E) method. The liver tissue was analyzed using ImageJ software vision 1.54f 29 June 2023 (NIH, Bethesda, MD, USA) (https://imagej.nih.gov/ij/, accessed on 10 August 2023) under a 100× light microscope with a Nikon Eclipse 80i digital camera (Nikon, Tokyo, Japan). 

### 4.4. RNA Isolation and Quantitative Real-Time PCR (qRT-PCR)

Liver tissue was used for gene expression analysis. Total RNA was extracted from the liver tissues using TRIzol reagent (Invitrogen, Carlsbad, CA, USA) and its concentration was assessed using a Nanodrop 2000c spectrophotometer (Thermo Fisher Scientific, Waltham, MA, USA). Complementary DNA (cDNA) was generated by reverse transcription of the total RNA using the primeScript RT reagent Kit (Takara, Otsu, Japan) following the manufacturer’s instructions. Subsequently, quantitative real-time polymerase chain reaction (qRT-PCR) analysis was performed using the SYBR premix Ex Taq Kit (Takara, Otsu, Japan). The samples were analyzed in triplicate. The mRNA expression levels were normalized using β-actin, and the relative gene expression was determined using the 2^−∆∆Ct^) method. The primers were designed in Table 2 as follows (presented in the sequence 5′-3′ forward, 5′-3′ reverse):

### 4.5. Cell Culture

The HepG2 cells used in this study were obtained from the ATCC cell bank. After resuscitation, they were cultured in a Thermo Fisher Scientific incubator (model 51033775, Waltham, MA, USA) at 37 °C with a 5% CO_2_ atmosphere. The culture medium used was IMDM (Gibco, Grand Island, NY, USA) supplemented with 10% fetal bovine serum (Gibco, Grand Island, NY, USA) and 1% penicillin/streptomycin (Procell, Wuhan, China).

Cell passaging was performed when the cell density reached 80–90%. The cells used in the experiments were from the 5th to 15th generations. HepG2 cells were cultured in a medium enriched with higher levels of fatty acids (palmitic acid/sodium oleate, 500 µM/250 µM) and glucose (50 mM). The sodium oleate compounds used were obtained from Sigma-Aldrich (Darmstadt, Germany) with catalog numbers O7501 and P9767. The glucose used was also obtained from Sigma-Aldrich (Germany) with catalog number D9434.

### 4.6. Immunofluorescence Staining

The cells on the culture plate were washed three times with PBS. The slides were then fixed with 4% paraformaldehyde for 15 min. For the paraffin sections, they were dewaxed using xylene (Sigma-Aldrich, Germany, catalog number 534056), rehydrated with ethanol (Sigma-Aldrich, Germany, catalog number 8.18760), and subjected to antigen retrieval using EDTA-Tris (Sigma-Aldrich, Germany, catalog number 93302). To prevent nonspecific binding, the slides were treated with goat serum (Gibco, Grand Island, NY, USA) and incubated at room temperature for 30 min.

Subsequently, a primary antibody was added to each slide and placed in a humidified chamber, where it was incubated overnight at 4 °C. Finally, a fluorescent secondary antibody was added and incubated in the humidified chamber at room temperature for 1 h. The slides were incubated with DAPI for 5 min, excess DAPI was then washed away, and the slides were sealed. Subsequently, the images were observed and captured under a fluorescence microscope.

The primary antibodies used in this study were BCL-2 (#15071, Cell Signaling Technologies, Danvers, MA, USA), PCNA (#13110, Cell Signaling Technologies, Danvers, MA, USA), and P65 (#8242, Cell Signaling Technologies, Danvers, MA, USA), diluted as follows. The secondary antibodies utilized were anti-rabbit IgG (H+L), F(ab’)2 Fragment (Alexa Fluor^®^ 488 Conjugate) (#4412, Cell Signaling Technologies, Danvers, MA, USA), and goat anti-rabbit IgG H&L (Cy3 ^®^) (ab97035, Abcam, Cambridge, MA, USA).

### 4.7. Western Blot Analysis

Preparation of lysate from tissues

Dissect the tissue of interest with clean tools, on ice preferably, and as quickly as possible to prevent degradation by proteases;Place the tissue in round-bottom microcentrifuge tubes or Eppendorf tubes and immerse in liquid nitrogen to snap freeze. For a ~5 mg piece of tissue, add ~300 μL of ice-cold lysis buffer rapidly to the tube, rinse the blade twice with another 2 × 200 μL lysis buffer;Centrifuge for 10 min at 14,000× *g* at 4 °C in a microcentrifuge.

Sample preparation

Remove a small volume of lysate to perform a protein quantification assay;Determine how much protein to load and add an equal volume 2× Laemmli sample buffer;Boil each lysate in sample buffer at 100 °C for 5 min.

Loading and running the gel

Load equal amounts of protein into the wells of the SDS-PAGE gel, along with a molecular weight marker. Load 20–30 μg of total protein from tissue homogenate;Run the gel for 1–2 h at 100 V.

Transferring the protein from the gel to the membrane

The membrane can be either nitrocellulose or PVDF. Activate PVDF with methanol for 1 min and rinse with transfer buffer before preparing the stack;Run for 120 min at 250 mAh.

After the blocking step, the membranes were incubated with the corresponding primary antibodies at 4 °C overnight. Subsequently, the membranes were treated with secondary antibodies at room temperature for 1 h. The PVDF membrane was washed, and the protein bands were visualized using the Bio-Rad ChemiDoc™ XRS system (Shanghai, China). The intensity of the bands was quantified using Image J software (https://imagej.nih.gov/ij/, accessed on 10 August 2023) [60]. The primary antibody dilutions used in this study were beta-actin (#4967, Cell Signaling Technologies, Danvers, MA, USA); caspase-3 (#9662, Cell Signaling Technologies, Danvers, MA, USA); ASC/TMS1 (#67824, Cell Signaling Technologies, Danvers, MA, USA); and NLRP3 (#15101, Cell Signaling Technologies, Danvers, MA, USA).

### 4.8. Cell Counting Kit 8 Assay

The concentration of kaempferol that yielded the best results was determined using the Cell Counting Kit-8 (CCK-8) (Dongren Chemical Technology Co., Ltd., Shanghai, China). The HepG2 cell line was seeded in 96-well plates and cultured for 24 h. The cells were then treated with different concentrations of kaempferol (10–50 μM), while the blank control group received an equivalent volume of cell-free medium. After incubating for 24 h, 10 μL of CCK-8 solution was added to each well and incubated in the dark for 2 h. The absorbance at 450 nm was measured using an enzyme labeling instrument (VLBLATGD2, Thermo Fisher Scientific, Waltham, MA, USA) [61,62].

### 4.9. Statistical Analysis

The data are presented as mean ± standard deviation (mean ± SD) of at least three independent experiments. Data analysis between two groups was performed using Kruskal–Wallis test. Data analysis and graphics drawing were performed using GraphPad Prism 10 (GraphPad Software, San Diego, CA, USA, www.graphpad.com). A *p*-value of less than 0.05 was considered statistically significant.

### 4.10. Ethical Statement

C57BL/6 female mice were purchased from Vital River Laboratory Animal Technology (Beijing, China). The animals were regularly maintained in the Medical School of University of Chinese Academy of Sciences. All the protocols involving the use of animals were in accordance with approved guidelines of the Institutional Review Board (or Ethics Committee) of the Institutional Animal Care and Use Committee (20230620-01 and 10 March 2023).

## 5. Conclusions

In summary, this study confirms that Kae can improve liver damage by regulating NLRP3-ASC/TMS1-Caspase 3 both in vivo and in vitro. Additionally, the expression of inflammatory factors is reduced. The role of Kae provides a new direction for the treatment of NASH, and NLRP3 can be a potential target for NASH treatment.

## Figures and Tables

**Figure 1 molecules-29-02630-f001:**
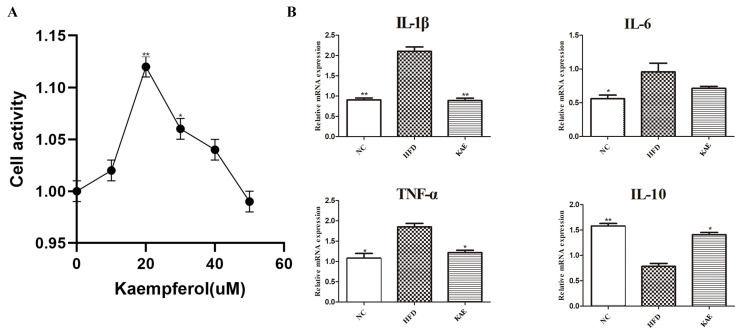
The Impact of kaempferol on HepG2 Cells. (**A**) CCK 8 analysis of HepG2 cell proliferation and the concentration gradient of Kae including: 10, 20, 30, 40, and 50 µM (compare with blank group). (**B**) The mRNA expression levels of IL-1β, IL-6, TNF α, and IL-10 in HepG2 cells were measured by qRT-PCR. Data are the mean ± SD. * *p* < 0.05 and ** *p* < 0.01 compared to the HFD group.

**Figure 2 molecules-29-02630-f002:**
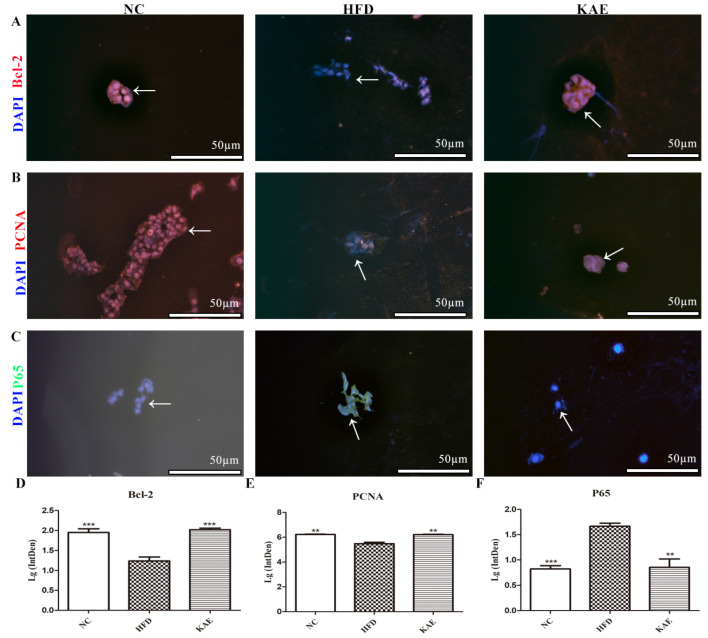
Immunofluorescence assay for assessing proliferative function in HepG2 cell line following kaempferol treatment. Immunofluorescence staining was analyzed for (**A**) BCL-2, (**B**) PCNA, and (**C**) P65 (NF-κB) protein expression in HepG2, which were all located in nucleus (200×), (**D**) average fluorescence intensity of Bcl-2, (**E**) average fluorescence intensity of PCNA, (**F**) average fluorescence intensity of P65. Data were expressed as mean ± standard deviation (SD) values. Average fluorescence intensity detection was used to detect proteins expression level and every protein was observed over 3 fields. ** *p* < 0.01, and *** *p* < 0.001 versus the HFD group. The white arrow represents the position of the significant difference.

**Figure 3 molecules-29-02630-f003:**
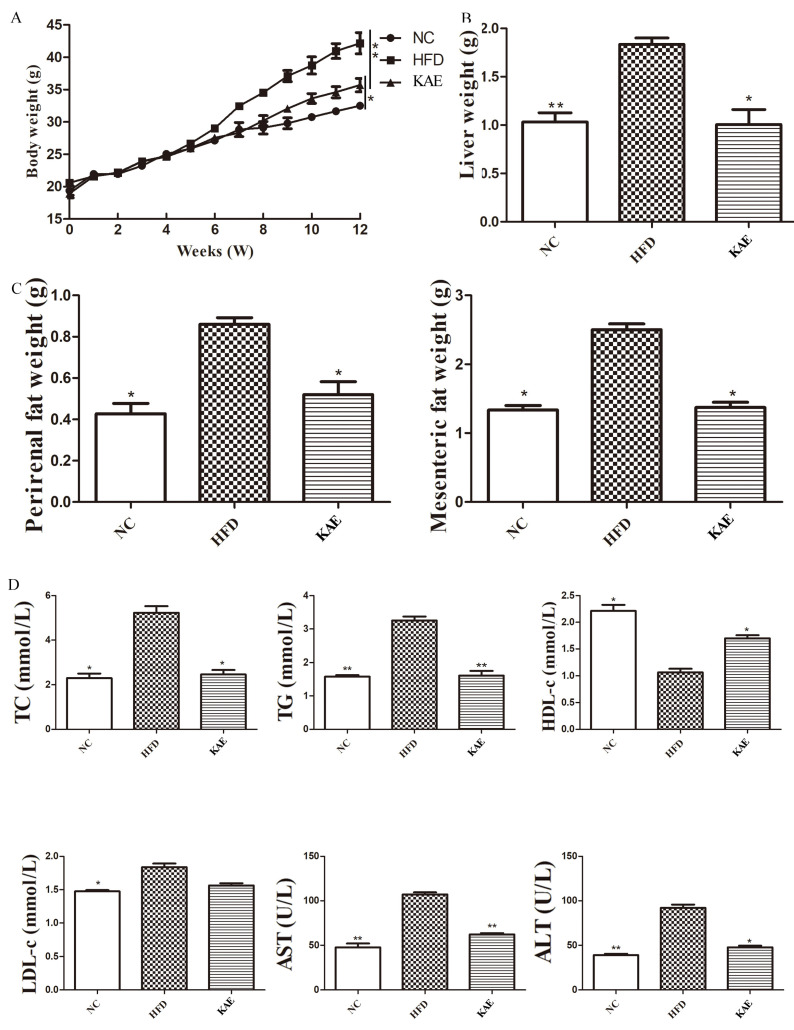
The effect of kaempferol on lipid metabolism in NASH model mice. (**A**) The body weight; (**B**) liver weight; (**C**) perirenal and mesenteric fat weight; (**D**) serum chemical biomarker including TC, TG, HDL, LDL, AST, and ALT. Data were expressed as mean ± standard deviation (SD) values. * *p* < 0.05, ** *p* < 0.01, and. versus the HFD group.

**Figure 4 molecules-29-02630-f004:**
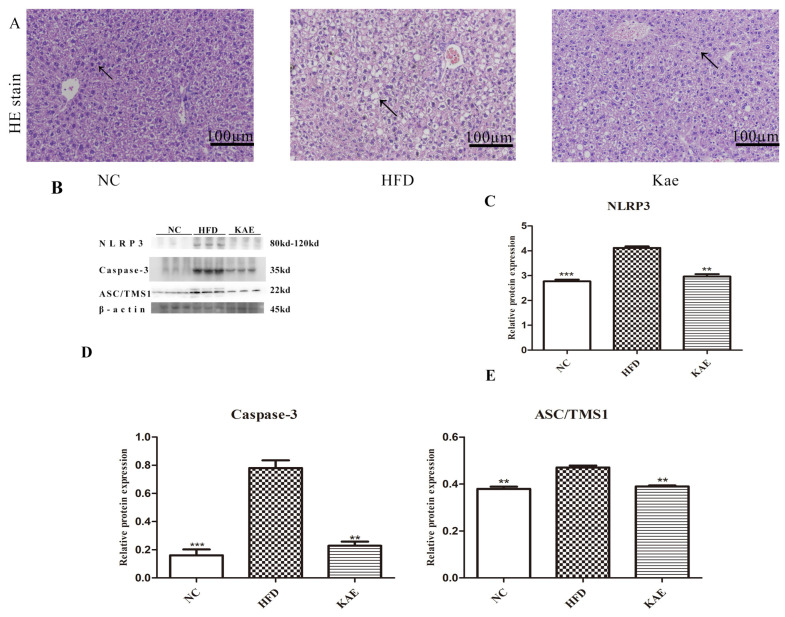
The NLRP3-ASC/TMS1/Caspase-3 pathway initiated liver injury in NASH mice. (**A**) HE analysis of liver tissue (100×). (**B**) Western blot results of NLRP3-ASC/TMS1-Caspase 3. (**C**–**E**) Quantitative analysis of Western blot image for NLRP3-ASC/TMS1-Caspase 3. Data are the mean ± SD. Compare with HFD group, ** *p* < 0.01, and *** *p* < 0. The black arrow represents the position of the significant difference.

**Figure 5 molecules-29-02630-f005:**
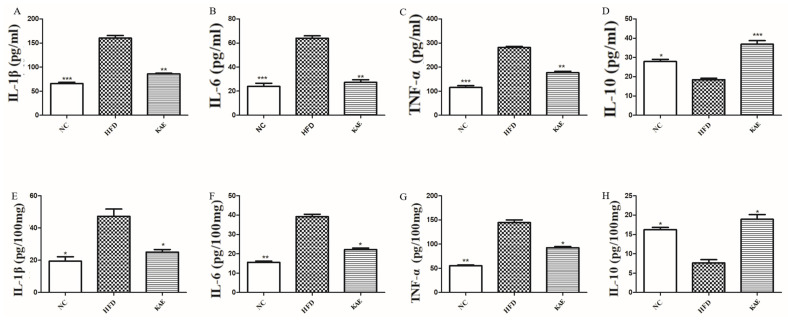
The function of kaempferol supplements on inflammation in a NASH mouse model: (**A**) IL-1β, (**B**) IL-6, (**C**) TNF-α, and (**D**) IL-10; liver tissue inflammatory factors: (**E**) IL-1β, (**F**) IL-6, (**G**) TNF-α, and (**H**) IL-10. Data were expressed as mean ± standard deviation (SD) values. * *p* < 0.05, ** *p* < 0.01, and *** *p* < 0. versus the HFD group.

**Figure 6 molecules-29-02630-f006:**
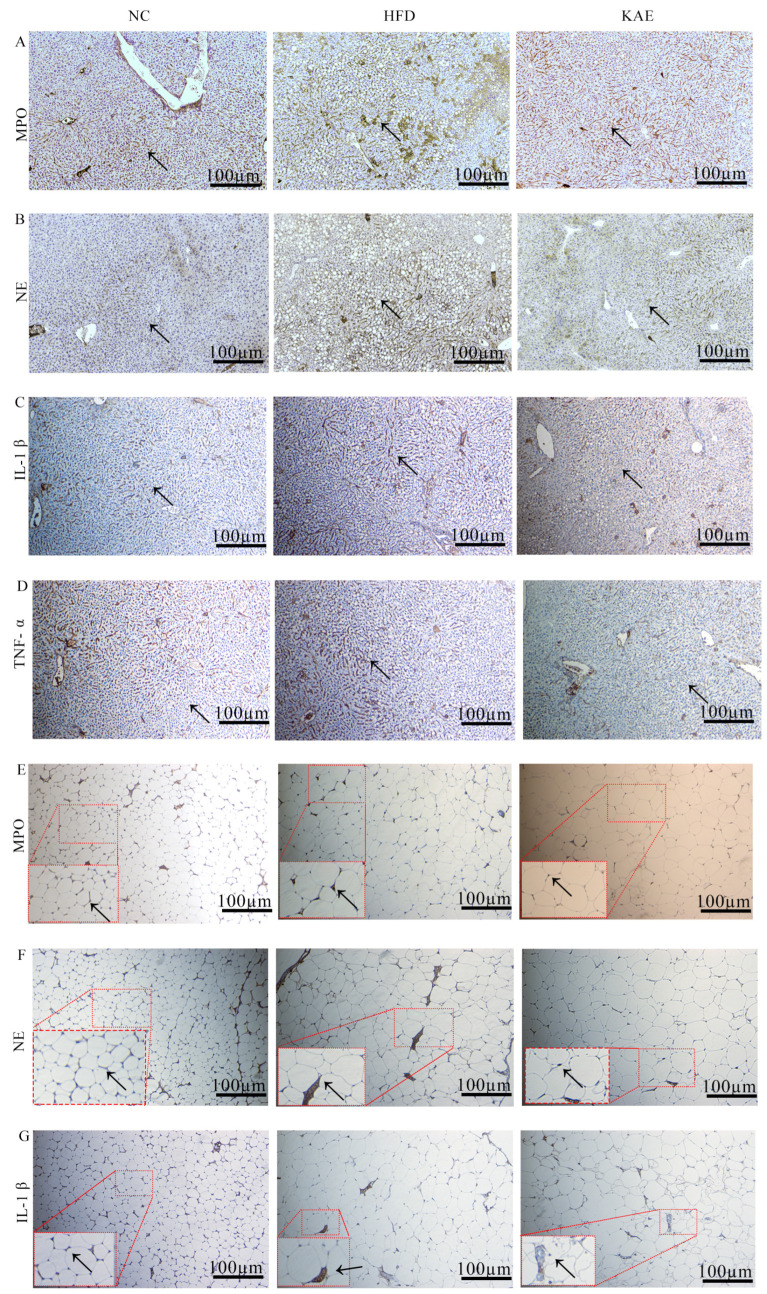
The activation of MPO and NE overexpression caused injury in NASH mice (100×). (**A**) MPO, (**B**) NE, (**C**) IL-1β, (**D**) TNF-α protein expression in liver; (**E**) MPO, (**F**) NE, (**G**) IL-1β expression in perivisceral adipose tissue. The black arrow represents the position of the significant difference.

**Figure 7 molecules-29-02630-f007:**
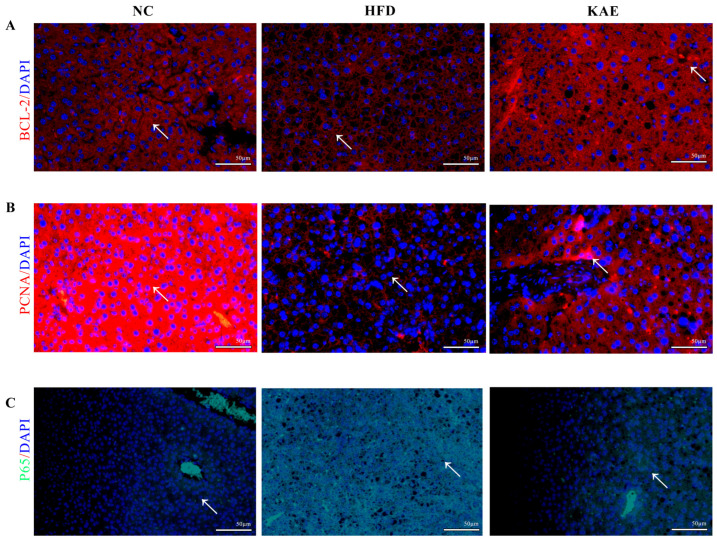
Effect of kaempferol on proliferation in liver tissue of NASH model mice. Effect of Kae on proliferation, anti-apoptosis, and anti-inflammation in liver tissue, where proteins were located in cell nucleus. Immunofluorescence staining was analyzed for (**A**) BCL-2, (**B**) PCNA, and (**C**) P65 (NF-κB) protein expression in liver tissue (200×). Every protein was observed over 3 fields. The white arrow represents the position of the significant difference.

**Table 1 molecules-29-02630-t001:** High-fat feed diet and nature control feed diet every 100-kcal energy constriction of fat, protein, and carbohydrate.

	High-Fat Diet with 60% Fat Energy	Nature Control Diet
Fat	60%	10%
Protein	20%	20%
Carbohydrate	20%	70%

**Table 2 molecules-29-02630-t002:** Primers details.

	NM Accession Code	5′-3′	3′-5′	Amplicon Length
IL-6	NM_012589.2	GCCTTCTTGGGACTGATGCT	TGTGACTCCAGCTTATCTCTTGG	199
IL-1β	NM_008361.4	TGCCACCTTTTGACAGTGATG	TTCTTGTGACCCTGAGCGAC	94
IL-10	NM_012854.2	ACTACCAAAGCCACAAGGCA	ACACCTTGGTCTGGAGCTTATTA	185
IL-4	NM_201270.1	TCACAGCAACGAAGAACACCA	CAGGCATCGAAAAGCCCGAA	172
β-actin	NM_004882	AAGGAGCCCCACGAGAAAAAT	ACCGAACTTGCATTGATTCCAG	169

## Data Availability

The datasets for this study are available from the corresponding author: gaoguolan@ucas.ac.cn.

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
