# Peer review of "Kaempferol Alleviates Hepatic Injury in Nonalcoholic Steatohepatitis (NASH) by Suppressing Neutrophil-Mediated NLRP3-ASC/TMS1-Caspase 3 Signaling"

_molecules, 2024, doi:10.3390/molecules29112630_

Round 1

Reviewer 1 Report

Comments and Suggestions for Authors

Dear authors,

The submitted manuscript presents the research aimed at clarification of mechanisms by which kaempferol exhibits beneficial effects in NASH. The study is well experimentally designed, the obtained results are of high scientific interest and the conclusions made are supported by the results. However, some manuscript modifications have to be implemented prior publication.

-in general, I would recommend proofreading of the manuscript by authors in order to correct typographic errors.  Moreover, please be advised to correct parts of sentences ending with references (I have highlighted some of the errors in yellow - in the attached file, but not all are highlighted; in most of the cases you should add space before stating reference in bracketes, i.e. "SPACE[REF]".

-do not use term "bioflavonoid", but simply "flavonoid"; or if You want to be more precise, in case of kaempferol, it is flavonoid aglycone, but "flavonoid" is good enough.

-do not start the sentence with abbreviation, specifically Kae

-a section ranging from line 76-89 is written in lyrical style, especially sentences from line 78-80, and 82-84; Please rephrase this section/these sentences to sound scientifically.

-section 4.1., line 352 - it states that 27 animals were divided in FOUR equal groups, each consisting of nine animals; I suppose that it should state THREE equal groups - please revise

-section 4.8. - The authors state using Student's t-test and one-way ANOVA in statistical processing of the obtained results, whereas the differences between two groups were assessed by Student's t-test, and between three groups by one-way ANOVA. This is not a good approach since multiple Student's t-tests should not be performed at all (in this case) for the purpose of post-hoc tests as this leads to an error, but one way ANOVA should be used for testing the differences and followed by appropriate ANOVA post-hoc test (i.e Tukey's HSD).

However, since the studied animal groups contained only 9 animals, and no data are available regarding the number of repeated measurements, the statistical processing in this study should be performed by application of ANOVA corresponding non parametric technique - Kruskal-Wallis test (followed by appropriate post-hoc test, ie. Dunn's test.) If, for any reason differences between two groups are supposed to be tested, than  Mann Whitney U test should be used instead of Student's t-test, but I do not think that the authors will require application of this technique.

In conclusion, I would add the suggestion that statistically significant differences between evaluated groups in studied parameters should be marked on graphs present in the manuscript. Moreover the graphs (not the images) should be of better quality with larger fonts, in order to increase the readability.

Kind regards.

Comments on the Quality of English Language

Minor editing required.

Author Response

(More detalis please check in the Word)

Response to review 1 comments

  1. Summary

Thank you very much for taking the time to review this manuscript. Please find the detailed responses below and the corresponding revisions highlighted changes in the re-submitted files.

  1.  

Quality of English Language

Minor editing of English language required

Does the introduction provide sufficient background and include all relevant references?

Yes

Are all the cited references relevant to the research?

Yes

Is the research design appropriate?

Yes

Are the methods adequately described?

Yes

Are the results clearly presented?

Can be improved

Are the conclusions supported by the results?

Yes

  1. Point-by-point response to Comments and Suggestions for Authors

Comment 1 :-in general, I would recommend proofreading of the manuscript by authors in order to correct typographic errors.  Moreover, please be advised to correct parts of sentences ending with references (I have highlighted some of the errors in yellow - in the attached file, but not all are highlighted; in most of the cases you should add space before stating reference in bracketes, i.e. "SPACE[REF]".

Response 1: Thanks for your review, in this revised version, we focused on correct typographic errors, and try our best to rectify. Please check our new manuscript with review revised model.

Comment 2: -do not use term "bioflavonoid", but simply "flavonoid"; or if You want to be more precise, in case of kaempferol, it is flavonoid aglycone, but "flavonoid" is good enough.

Response 2: Agree with your suggestion, we use "flavonoid" to replace "bioflavonoid".

Comment 3: -do not start the sentence with abbreviation, specifically Kae.

Response 3: We are sorry for that, and we consulted previous studies that referred to kaempferol as Kae for short, so we followed this abbreviation.

Comment 4:-a section ranging from line 76-89 is written in lyrical style, especially sentences from line 78-80, and 82-84; Please rephrase this section/these sentences to sound scientifically.

Response 4: Thank you for pointing this out. We agree with this comment. Therefore, we have remodified this section as: And Kae's prowess in mitigating the risk of liver failure induced by the harmful lipo-polysaccharide-galactosamine, achieved through the moderation of the deleterious en-doplasmic reticulum stress (ERS). (line 119)

Comment 5: -section 4.1., line 352 - it states that 27 animals were divided in FOUR equal groups, each consisting of nine animals; I suppose that it should state THREE equal groups - please revise

Response 5: We are sorry for the mistake, we have check the groups, there were three groups, and amend in the manuscript. (line 907)

Comment 6: -section 4.8. - The authors state using Student's t-test and one-way ANOVA in statistical processing of the obtained results, whereas the differences between two groups were assessed by Student's t-test, and between three groups by one-way ANOVA. This is not a good approach since multiple Student's t-tests should not be performed at all (in this case) for the purpose of post-hoc tests as this leads to an error, but one way ANOVA should be used for testing the differences and followed by appropriate ANOVA post-hoc test (i.e Tukey's HSD).

However, since the studied animal groups contained only 9 animals, and no data are available regarding the number of repeated measurements, the statistical processing in this study should be performed by application of ANOVA corresponding non parametric technique - Kruskal-Wallis test (followed by appropriate post-hoc test, ie. Dunn's test.) If, for any reason differences between two groups are supposed to be tested, than  Mann Whitney U test should be used instead of Student's t-test, but I do not think that the authors will require application of this technique.

Response 6: Thank you for pointing this out. And we have retested the results with Kruskal-Wallis test and recorrected those in ever figures and results. The details were showed in the manuscripts, thanks for your supervision.

Comment 7: In conclusion, I would add the suggestion that statistically significant differences between evaluated groups in studied parameters should be marked on graphs present in the manuscript. Moreover the graphs (not the images) should be of better quality with larger fonts, in order to increase the readability.

Response 7: Thanks for your suggestion, we have enlarged the fonts in the graphs for easier to read.

In the end, lots of thanks for you for your review.

  1. Response to Comments on the Quality of English Language

Point 1: Minor editing required.

Response 1: We were sorry for that. If the quality of English language keeps any discomfort for you to read, we have look for help by professional English writer and hope this vision could be easier for review.

Yours Sincerely,

Guolan Gao

18 May 2024

University of Chinese Academy of Sciences

gaoguolan@ucas.ac.cn

Reviewer 2 Report

Comments and Suggestions for Authors

The manuscript describes some of the possible pathways involved in the hepatoprotective effect of Kaempferol (Kae) in a model of nonalcoholic steatohepatitis (NASH) using mice. I recognize the merit in performing such work since NASH is a prevalent hepatic condition and Kae is a well-known natural compound associated with many biological effects. However, overall, the manuscript needs substantive changes. The manuscript lacks clarity, especially in the presentation of the results. The work has several major weaknesses that must be addressed. Please find below the considerations made for the manuscript.

1) The western blot images are not acceptable and better figures must be provided by the authors.

2) The results are difficult to visualize. The layout of the graphs and figures needs to be standardized. Generally, the quality of the figures needs improvement. Histological and immunofluorescence images should clearly depict the differences between groups with annotations such as arrows or asterisks.

3) Standardize the use of abbreviations and acronyms in the text. They should be mentioned in the first appearance and then used consistently throughout the text.

4) The vocabulary, language, and terms used in some parts of the text, mostly in the Introduction, are not adequate and must be improved. Although the authors intended to add some style to their writing, the terminology used does not meet the requirements of a research paper. The Introduction is too long; I suggest a considerable reformulation of this section. I even recommend the authors delete some sentences and rephrase some sections that do not add to the text, such as in paragraph 3 (L. 76-90) of the Introduction. Generally, this paragraph does not contribute to the Introduction.

5) In line with the previous comment, many terms are used throughout the text that should be revised, rephrased, or deleted. Please see some examples: tempering of the tempestuous (L. 84); innocent (L. 88); majesty (L. 89); gracefully (L. 236, L. 325); gives birth (L. 269); talisman (L. 273); sacred bonds (L. 276); unleashes the hallowed (L. 280); ushering in the sacrament (L. 282); celestial realm (L. 319); alchemist of harmony (L. 323-324); and many others.

There is also a repetition of words that make the text unattractive. For example, the words tapestry, orchestrates/orchestrating, symphony, intricate, and realm, are repeated several times.

6) What do the authors mean by “The precise subject-specific vocabulary is used where necessary. Cellular models provide additional insights.”? (L. 310-311).

7) Why did the authors perform their study with female mice? This needs to be explained.

8) The dose/concentration of Kaempferol (Kae) used for in vitro and in vivo protocols should be informed in the abstract.

9) In the Discussion (L. 231), did the authors mean Kaempferia galanga L.? Also, the scientific names of species should be italicized.

10) To allow the reproducibility of the experiments, the methods should be described in more detail even if published before.

11) Section 4.1 “Animals and Experimental Design” – In this part “27 mice were randomly assigned to four groups, each consisting of 9 mice: (1) the nature control (NC) group, (2) the higher fatty acid-fed group (HFD) (details are shown in table 1), (3) the higher fatty acid with Kaempferol-fed group.” (L. 352-354), the information about the experimental number of animals in each group is incorrect. The graphs only show 3 groups. Please check.

12) Section 4.3 “RNA Isolation and Quantitative Real-Time PCR (qRT-PCR)” – In this part “Liver tissue was used for gene expression analysis. Total RNA was extracted from the vaginal tissues using TRIzol reagent” (L. 386-387), why do authors mention vaginal tissues? For better visualization, provide primer sequences in a table. Also, include the Forward and Reverse sequences for β-actin. The NM accession code and amplicon length should also be provided for each primer.

13) Section 4.6 “Western Blot Analysis” – How were samples homogenized and prepared for western blot analysis, running, and transfer conditions? Provide this information.

14) Section 4.7 “Cell Counting Kit 8 Assay” – Is there a reason/reference to support the different concentrations of Kaempferol (10-50 μM) used in this study for the in vitro protocol?

Also, there is no coherence between in vitro and in vivo experiments.

The protocols performed with the liver tissue should also be done with the HepG2 cell line, such as the measurements of inflammatory cytokines and NLRP3/Caspase-3 signaling.

15) The English needs to be revised to improve the quality of the manuscript. Some parts are difficult to read. I suggest that the authors seek the assistance of a native speaker to polish the text.

16) Revise the Institutional Review Board Statement (L. 480-482).

Comments on the Quality of English Language

The English needs to be revised to improve the quality of the manuscript. Some parts are difficult to read. I suggest that the authors seek the assistance of a native speaker to polish the text.

Author Response

(More details please check in the Word)

Response to review 2 comments

  1. Summary

We feel great thanks for your professional review work on our article. As you are concerned, there are several problems that need to be addressed. According to your nice suggestions, we have made extensive corrections to our previous draft, the detailed corrections are listed below.

  1.  

Quality of English Language

Moderate editing of English language required

Does the introduction provide sufficient background and include all relevant references?

Must be improved

Are all the cited references relevant to the research?

Can be improved

Is the research design appropriate?

Must be improved

Are the methods adequately described?

Can be improved

Are the results clearly presented?

Must be improved

Are the conclusions supported by the results?

Can be improved

  1. Point-by-point response to Comments and Suggestions for Authors

Comment 1) The western blot images are not acceptable and better figures must be provided by the authors.

Response 1: Thank you very much for pointing out this important issue. We agree with your opinion. Unfortunately, due to the limited time and the rare samples, we did not supplement experimental validation. But we improve the quality of the WB images. We hope it could make up for it.

Comment 2) The results are difficult to visualize. The layout of the graphs and figures needs to be standardized. Generally, the quality of the figures needs improvement. Histological and immunofluorescence images should clearly depict the differences between groups with annotations such as arrows or asterisks.

Response 2: We gratefully appreciate for your valuable suggestion. In the revised vision, we improve the quality of graphs and figures. More details please check in manuscript. Hope it will be easier for reading.

Comment 3) Standardize the use of abbreviations and acronyms in the text. They should be mentioned in the first appearance and then used consistently throughout the text.

Response 3: Thanks for your careful checks. We are sorry for our carelessness. Based on your comments, we have made the corrections to make the word harmonized within the whole manuscript.

Comment 4) The vocabulary, language, and terms used in some parts of the text, mostly in the Introduction, are not adequate and must be improved. Although the authors intended to add some style to their writing, the terminology used does not meet the requirements of a research paper. The Introduction is too long; I suggest a considerable reformulation of this section. I even recommend the authors delete some sentences and rephrase some sections that do not add to the text, such as in paragraph 3 (L. 76-90) of the Introduction. Generally, this paragraph does not contribute to the Introduction.

Response 4: We sincerely appreciate the valuable comments. We have re-written this part according to the Reviewer's suggestion. And we rewrite the whole manuscript hope it could be helpful for reading.

Comment 5) In line with the previous comment, many terms are used throughout the text that should be revised, rephrased, or deleted. Please see some examples: tempering of the tempestuous (L. 84); innocent (L. 88); majesty (L. 89); gracefully (L. 236, L. 325); gives birth (L. 269); talisman (L. 273); sacred bonds (L. 276); unleashes the hallowed (L. 280); ushering in the sacrament (L. 282); celestial realm (L. 319); alchemist of harmony (L. 323-324); and many others.

There is also a repetition of words that make the text unattractive. For example, the words tapestry, orchestrates/orchestrating, symphony, intricate, and realm, are repeated several times.

Response 5: Thank you again for your positive and constructive comments, as the fellow we have rewritten the whole manuscript please review our new vision of manuscript.

Comment 6) What do the authors mean by “The precise subject-specific vocabulary is used where necessary. Cellular models provide additional insights.”? (L. 310-311).

Response 6: We feel sorry for our carelessness. In our resubmitted manuscript, the mistake of reference citing is revised. Thanks for your correction. (Cellular models have shed light on Kae's regulatory role in apoptosis, contributing to liver homeostasis restoration. This finding holds promise for developing innovative treatments for NASH and other inflammatory liver diseases. ( Line.741-743)

Comment 7) Why did the authors perform their study with female mice? This needs to be explained.

Response 7: Thank you for your comment. One part of our study was a retrospective analysis of clinical patients, and the other part was an animal study. And according to our clinical data found menopausal women are suffering from metabolic disorder syndrome related function, so we choose as our model animals, older females.

Comment 8) The dose/concentration of Kaempferol (Kae) used for in vitro and in vivo protocols should be informed in the abstract.

Response 8: Thanks for you valuable suggestion, and we added the details in abstract as fellow. Kaempferol (Kae) was investigated for its ability to modulate systemic inflammatory responses and lipid metabolism in this model (20 mg/kg per day). Notably, Kae significantly reduced the expression of NLRP3-ASC/TMS1-Caspase 3, a crucial mediator of liver tissue inflammation. Additionally, in a HepG2 cell model induced with palmitic acid/oleic acid (PA/OA) to mimic NASH conditions, Kae demonstrated the capacity to decrease lipid droplet accumulation and downregulate the expression of NLRP3-ASC/TMS1-Caspase 3(20µM and the final concentration was 20 nM). Line 15-20.

Comment 9) In the Discussion (L. 231), did the authors mean Kaempferia galanga L.? Also, the scientific names of species should be italicized.

Response 9: Thanks for your careful checks. We have re-written this part according to the Reviewer's suggestion. Kaempferol (Kae), a naturally occurring flavonoid found in various fruits and veg-etables, including Kaempferol galanga L., possesses a remarkable range of health-promoting properties. (Line 539)

Comment 10) To allow the reproducibility of the experiments, the methods should be described in more detail even if published before.

Response 10: We think this is an excellent suggestion. We have rewritten the methods part and please review the new vision resubmitted.

Comment 11) Section 4.1 “Animals and Experimental Design” – In this part “27 mice were randomly assigned to four groups, each consisting of 9 mice: (1) the nature control (NC) group, (2) the higher fatty acid-fed group (HFD) (details are shown in table 1), (3) the higher fatty acid with Kaempferol-fed group.” (L. 352-354), the information about the experimental number of animals in each group is incorrect. The graphs only show 3 groups. Please check.

Response 11: We were sorry for our careless mistakes. Thank you for your reminder. There were three groups and fixed in the new type. (Line 907)

Comment 12) Section 4.3 “RNA Isolation and Quantitative Real-Time PCR (qRT-PCR)” – In this part “Liver tissue was used for gene expression analysis. Total RNA was extracted from the vaginal tissues using TRIzol reagent” (L. 386-387), why do authors mention vaginal tissues? For better visualization, provide primer sequences in a table. Also, include the Forward and Reverse sequences for β-actin. The NM accession code and amplicon length should also be provided for each primer.

Response 12: We feel sorry for our carelessness. Due to the research and the manuscript were cooperation with several. And every one response for they own part. So, another study was mistakenly included in this manuscript, which was about vaginal tissue. And the part was checked and provide more detail as fellow.

NM accession code

5`-3`

3`-5`

amplicon length

IL-6

NM_012589.2

GCCTTCTTGGGACTGATGCT

TGTGACTCCAGCTTATCTCTTGG

199

IL-1β

NM_008361.4

TGCCACCTTTTGACAGTGATG

TTCTTGTGACCCTGAGCGAC

94

IL-10

NM_012854.2

ACTACCAAAGCCACAAGGCA

ACACCTTGGTCTGGAGCTTATTA

185

IL-4

NM_201270.1 

TCACAGCAACGAAGAACACCA

CAGGCATCGAAAAGCCCGAA

172

β-actin

NM_004882

AAGGAGCCCCACGAGAAAAAT

ACCGAACTTGCATTGATTCCAG

169

Table 2 : Primers details.

Comment 13) Section 4.6 “Western Blot Analysis” – How were samples homogenized and prepared for western blot analysis, running, and transfer conditions? Provide this information.

Response 13: As suggested by the reviewer, we have added more details to support this process as fellow: (Line 479-510)

4.7. Western Blot Analysis

Preparation of lysate from tissues

  1. Dissect the tissue of interest with clean tools, on ice preferably, and as quickly as possible to prevent degradation by proteases.
  2. Place the tissue in round-bottom microcentrifuge tubes or Eppendorf tubes and immerse in liquid nitrogen to snap freeze. For a ~5 mg piece of tissue, add ~300 μL of ice-cold lysis buffer rapidly to the tube, rinse the blade twice with another 2 x 200 μL lysis buffer.
  3. Centrifuge for 10 min at 14,000 g at 4°C in a microcentrifuge.

Sample preparation

  1. Remove a small volume of lysate to perform a protein quantification assay.
  2. Determine how much protein to load and add an equal volume 2X Laemmli sample buffer.
  3. Boil each lysate in sample buffer at 100°C for 5 min.

Loading and running the gel

  1. Load equal amounts of protein into the wells of the SDS-PAGE gel, along with a molecular weight marker. Load 20–30 μg of total protein from tissue homogenate.
  2. Run the gel for 1–2 h at 100 V.

Transferring the protein from the gel to the membrane

  1. The membrane can be either nitrocellulose or PVDF. Activate PVDF with methanol for 1 min and rinse with transfer buffer before preparing the stack.
  2. Run for 120 min at 250 mAh.

After the blocking step, the membranes were incubated with the corresponding primary antibodies at 4°C overnight. Subsequently, the membranes were treated with secondary antibodies at room temperature for 1 hour. The PVDF membrane was washed, and the protein bands were visualized using the Bio-Rad ChemiDoc™ XRS system (Shanghai, China). The intensity of the bands was quantified using Image J software (https://imagej.nih.gov/ij/, accessed on 10 August 2023). The primary antibody dilu-tions used in this study were: beita-actin(#4967, Cell Signaling Technologies, Danvers, Massachusetts, USA); caspase-3(#9662, Cell Signaling Technologies, Danvers, Massa-chusetts, USA); ASC/TMS1(#67824, Cell Signaling Technologies, Danvers, Massachusetts, USA); NLRP3(#15101, Cell Signaling Technologies, Danvers, Massachusetts, USA).

Comment 14) Section 4.7 “Cell Counting Kit 8 Assay” – Is there a reason/reference to support the different concentrations of Kaempferol (10-50 μM) used in this study for the in vitro protocol?

Also, there is no coherence between in vitro and in vivo experiments.

The protocols performed with the liver tissue should also be done with the HepG2 cell line, such as the measurements of inflammatory cytokines and NLRP3/Caspase-3 signaling.

Response 14: We sincerely appreciate the valuable comments. We have checked the literature carefully and added more references on reference to support the different concentrations of Kaempferol (10-50 μM) used in this study into the part in the revised manuscript, please reviews in our new vision resubmitted. And for the coherence, thank you very much for pointing out this important issue. We agree with your opinion. In this study, we aimed to explore the molecular mechanism of gene overexpression, and due to the mitochondria as the central part of adiposis and ROS reaction is the key point to induced it, so we tested the ROS index in the cell model to response the pressure in cell to reflect the tissue condition.

Comment 15) The English needs to be revised to improve the quality of the manuscript. Some parts are difficult to read. I suggest that the authors seek the assistance of a native speaker to polish the text.

Response 15: Thanks for your suggestion. We have tried our best to polish the language in the revised manuscript.

Comment 16) Revise the Institutional Review Board Statement (L. 480-482).

Response 16: We have re-written this part according to the Reviewer's suggestion. And the new vision as : Institutional Review Board Statement: The study protocol was approved by the Medical School of University of Chinese Academy of Sciences Code: 20230620-01. (Line 1184-1185)

  1. Response to Comments on the Quality of English Language

Point 1: The English needs to be revised to improve the quality of the manuscript. Some parts are difficult to read. I suggest that the authors seek the assistance of a native speaker to polish the text.

Response 1: We tried our best to improve the manuscript and made some changes to the manuscript. These changes will not influence the content and framework of the paper. And here we did not list the changes but marked in red in the revised paper. We appreciate for reviewers’ warm work earnestly and hope that the correction will meet with approval.

Yours Sincerely,

Guolan Gao

18 May 2024

University of Chinese Academy of Sciences

gaoguolan@ucas.ac.cn

Round 2

Reviewer 2 Report

Comments and Suggestions for Authors

Dear Authors,

Thank you for submitting the revised version of the manuscript.

However, I still believe there are some areas that could be improved upon.

1. The layout of the graphs and figures still needs to be standardized. For example, the layout between graphs in Figure 1B and those in Figure 2D-F is inconsistent.

2. Is the sentence in lines 72-73, “It is important to use clear and objective language when discussing their properties [22].” truly necessary? In the Discussion section, please consider revising the sentences in lines 334-335, “In the realm of bioinformatics, the narrative of NAFLD unfolds like a tapestry of interconnected historical pathways”, and in lines 350-352, “Our journey will be guided by the pursuit of knowledge, as we seek to unravel the mysteries that lie within these pathways and their relationship with Kae”.

3. I suggest incorporating the response provided by the Authors for question 7 (female mice) into the text, possibly in the Discussion section. This information is an interesting detail about the methodology and may also serve as a constraint for the current study.

Thank you for your attention to these points.

Author Response

(For more details please check our Word vision)

Response to review 2 comments

  1. Summary

Thank you very much for taking the time to review this manuscript. Please find the detailed responses below and the corresponding revisions highlighted changes in the re-submitted files.

  1.  

Quality of English Language

English language fine. No issues detected

Does the introduction provide sufficient background and include all relevant references?

Yes

Are all the cited references relevant to the research?

Yes

Is the research design appropriate?

Yes

Are the methods adequately described?

Can be improved

Are the results clearly presented?

Yes

Are the conclusions supported by the results?

Yes

  1. Point-by-point response to Comments and Suggestions for Authors

Comment 1: The layout of the graphs and figures still needs to be standardized. For example, the layout between graphs in Figure 1B and those in Figure 2D-F is inconsistent.

Response 1: Thanks for your review, in this revised version, we have standardized the figures. Hope the new vision may be easier for you read.

Comment 2: Is the sentence in lines 72-73, “It is important to use clear and objective language when discussing their properties [22].” truly necessary? In the Discussion section, please consider revising the sentences in lines 334-335, “In the realm of bioinformatics, the narrative of NAFLD unfolds like a tapestry of interconnected historical pathways”, and in lines 350-352, “Our journey will be guided by the pursuit of knowledge, as we seek to unravel the mysteries that lie within these pathways and their relationship with Kae”.

Response 2: We are sorry for that, and we rewrite those sentences to more accuracy as fellow. “These compounds have the potential to alleviate a range of ailments. It is important to use clear and objective perspective when discussing their properties“(Line 110-111) and “In the field of bioinformatics, the function of NAFLD demonstrates more complex. “(Line 885).

Comment 3: I suggest incorporating the response provided by the Authors for question 7 (female mice) into the text, possibly in the Discussion section. This information is an interesting detail about the methodology and may also serve as a constraint for the current study.

Response 3: Thank you for pointing this out. We agree with this comment. Therefore, in discussion we keep this detail as “Because one part of our study was a retrospective analysis of clinical patients, and the other part was an animal study. And according to our clinical data found menopausal women are suffering from metabolic disorder syndrome related function, so we choose as our model animals, older females. “(Line 876-879).

Thank you very much for your attention and time. Look forward to hearing from you.

Yours Sincerely,

Guolan Gao

28 May 2024

University of Chinese Academy of Sciences

gaoguolan@ucas.ac.cn